# Learning Design Strategies in MOOCs for Physicians’ Training: A Scoping Review

**DOI:** 10.3390/ijerph192114247

**Published:** 2022-10-31

**Authors:** Giovanni Schettino, Vincenza Capone

**Affiliations:** Department of Humanities, University of Naples Federico II, 80133 Naples, Italy

**Keywords:** massive open online course, physicians, medical education, continuing education, eLearning, healthcare workers, review

## Abstract

In recent years, there has been an increased implementation of massive open online courses (MOOCs). This teaching model plays a pivotal role in online education because it can provide high-quality learning resources to numerous students with great feasibility, shaping training courses according to their different learning requirements. Although the widespread adoption of MOOCs in medical education has led to numerous benefits for undergraduate and graduate doctors, their role remains unclear, suggesting the need to analyze the key factors of such a learning method in this field. To achieve this aim, a scoping review, in line with the PRISMA method for qualitative synthesis, was performed by considering studies published from 2016 to 2021, written in English, and including the physician population. Through this literature analysis, the following main areas of interest came to light: (1) pedagogical approaches, (2) MOOC structure-related variables, (3) participant-related variables, and (4) MOOCs vs. traditional courses. The review provides valuable evidence on factors underlying MOOCs effectiveness, which might be helpful for academic and healthcare organizations in designing effective training courses for physicians.

## 1. Introduction

The European Commission [1] has identified specific priorities for the period 2021–2027 concerning the health field. More specifically, a primary point of Europe’s commitment is the improvement of health systems and the healthcare workforce by promoting their resources and enhancing access to quality, patient-centered, outcome-based healthcare and related care services.

In this regard, literature has demonstrated that the adoption of human resource development interventions can be helpful in increasing healthcare workers’ (HCWs) knowledge, skills, resilience, and quality of care [2], which is consistent with the human capital theory [3].

In the healthcare field, the need to ensure effective and economically sustainable training has led to a focus on technology-enhanced learning (TEL) and, specifically, to massive open online courses or MOOCs [4,5]. These courses can be regarded as a significant step forward in the educational field due to the ease of access and convenient content delivery they offer. More specifically, MOOCs are courses potentially aimed at an unlimited number of students (massive) who can attend them whenever and wherever they want. Moreover, they can register free of charge (open) and enjoy learning-oriented content accessible via the internet (online). Finally, the courses are characterized by a structured learning plan (course) developed by some of the world’s most prestigious universities [6].

The success of MOOCs is highlighted by the impressive number of individuals who have adopted them. These courses have reached over 220 million learners (40 million new learners in 2021 alone), with over 19.400 courses available and 1.416 covering the field of health and medicine [7]. The wide appeal of MOOCs in medical education can be explained by considering their potential both for HCWs and their organizations [8,9]. They represent an accessible and affordable choice by providing online learning in a more effective and timely manner compared with traditional education [10] since online learning has fewer time and space constraints [11]. Indeed, the quick implementation of this teaching mode has been proven to be particularly suitable for dealing with challenges concerning health emergencies by providing effective training for healthcare professionals [12,13]

Despite these advantages, scholars and developers of online courses have expressed skepticism about their ability to provide relevant education which advances people in careers and socio-economic activities [14]. These reservations are supported by the higher dropout levels in MOOCs than in offline courses [15]. It is widely acknowledged that the number of trainees drops by the end of courses, rarely reaching more than 7–9% of the initial students [16,17]. With this in mind, researchers have argued that the ability of MOOCs structure to “tune” to participants’ needs is crucial to limit this phenomenon and, in turn, realize their learning objectives [18,19,20,21]. Nevertheless, there is a scant amount of agreement about the definition of characteristics of a MOOC and its pedagogical foundations [5]. The reasons for this are twofold and are linked to the multifarious nature of this teaching mode [22]. The latter manifests itself in the extensive variations of the teaching model according to courses content, their aims, learners’ characteristics, and the innovations in TEL. Concerning the adoption of such a model in physicians’ training, few recent review studies [23,24] have systematically analyzed their effectiveness in this population, however, neglecting to investigate the factors that might explain these outcomes. In light of the above, the purpose of the current explorative study was to scrutinize the literature to extend the previous reviews on MOOCs for physicians by adding more updated sources; additionally, we investigated which were the key factors in building a MOOC that could effectively support these professionals in their clinical practice.

### The MOOC Model in the Healthcare Field

Although medicine is traditionally slow in adapting to new trends [24], the MOOCs revolution has strongly impacted this field. It must be noted that the widespread use of such a teaching method has been established in a relatively short time [25]. The first generation of MOOCs, or cMOOCS, are inspired by connectivism [26] due to their emphasis on social learning, cooperation, creativity, and knowledge sharing. The second generation of MOOCs, or xMOOCs, are based on a more traditional classroom structure centered around the teacher rather than a community of students. They provide limited communication space between participants [27]. Even though xMOOCs have been criticized for not providing adequate learner support and engagement, this teaching model has shown a high degree of flexibility and scalability [14]. These features have made it the primary teaching mode in online learning, contributing, in this way, to the broad adoption of MOOCs in various fields (e.g., engineering, math, law, social and healthcare sciences) and at different educational levels (e.g., primary, middle and high schools, undergraduate, postgraduate, and continuing education courses) [28]. More specifically, these online courses are capable of meeting organizations’ demands for properly trained employees since, as the human capital theory [29,30,31] posits, investing in employees’ training can lead to improvements in their performance. Valuable education and skills training contribute to an increase in both employees’ and organizations’ performances, evaluated in terms of outcomes such as job satisfaction, motivation, mental health, better standing, the development of new services, increased market share, and improved quality of services [32,33,34,35]. The latter is a critical success factor for field service organizations, since it depends mainly on customer satisfaction and the perceived value of the services rendered, as shown by Parasuraman et al. [36]. In this vein, employees’ training can manifest itself in the form of continuing education, which may be defined as “the process of engaging in educational pursuits with the goal of becoming up-to-date in the knowledge and skills of one’s profession” [37] (p. 4). This is the main dimension of human resources management practices due to its potential to capitalize the intangible assets of knowledge, skills, and abilities possessed by organizational actors, which, in turn, can translate into a competitive advantage in the market [38] and the achievement of organizations’ aims [39].

Regarding the teaching methods required to implement such a strategy, it is important to note that healthcare has been marked by the rise in TEL [40] fostered by steadily growing internet access and the wide adoption of web-based platforms in individual lives [41]. More precisely, massive open online courses or MOOCs have been the main subject of this interest.

As mentioned above, MOOCs provide effective, flexible, and economically sustainable training [42], responding to the need to improve the quality of care and, at the same time, improve cost efficiency [43]. As Mangalji and Karthikeyan [44] argue, the integration of MOOCs into medical training could contribute to improving students’ preparation, specifically through case-based learning, since it can lead them to focus on interactive aspects (e.g., problem-solving, team-based skills, and knowledge application). On that note, MOOCs that include active learning modules, feedback to learners, and the possibility to choose between different activities can foster students’ engagement and retention of information [44,45]. Moreover, MOOCs reported equal or better performance compared to traditional teaching [23,24]. The above-mentioned advantages related to such a teaching mode, mainly in terms of high flexibility, blend well with continuing education in the healthcare field since physicians require regular updates of the knowledge needed to be effective in the medical profession and deal adequately with health emergencies [12,13]. Hence, the adoption of the MOOC model represents a good way to provide continuing professional development [46] by combining physicians’ demand for high-quality training—in line with their time and resources—and the organizations’ efforts to achieve their goals.

In spite of their multiple benefits, MOOCs have several weaknesses, which translate into high dropout rates [47,48]. The latter can be explained by considering factors such as the easy registering process, high passiveness, low student motivation, and demands (including time) that are incompatible with participants’ private and professional duties, as well as the lack of necessary knowledge and skills to attend courses [17,49,50].

Following this reasoning, a MOOC should be built through the accurate selection of the most suitable parameters according to learners’ characteristics, although it is not easy in the presence of a considerable number of participants, which is typical of this teaching model. Thus, to develop an effective MOOC for physicians, it is necessary to gain adequate knowledge about the factors that contribute to the success of this teaching model among the medical population. In this regard, Milligan [51] (p. 1882) stated that “understanding the nature of learners and their engagement is critical to the success of any online education provision, especially those MOOCs where there is an expectation that the learners should self-motivate and self-direct their learning”.

In recent years, a large number of reviews have been conducted by considering studies on MOOCs related to healthcare workers in general [52,53,54], whereas, to the best of our knowledge, few studies have focused on physicians by paying attention solely to the outcomes of such courses. More precisely, Zhao et al. [24] performed a meta-analysis based on the examination of four articles reporting no significant differences between the efficacy of the MOOC teaching model and that of the traditional one. A subsequent meta-analysis study by Gao et al. [23], considering the same topic, reported that students enrolled in a MOOC performed better than those who attended traditional courses. However, it is important to note that this research included only studies at colleges or universities in China.

In light of this, the need to better understand the strategies and methods underlying the effectiveness of these courses is clear. Such recognition is fundamental for the passage from a theoretical to a practical level, offering, in this way, valuable help to developers in designing MOOCs for physicians that are capable of achieving their specific learning goals.

## 2. Methods

Due to the above-mentioned aim of the research, a scoping literature review seemed the most appropriate approach to adopt since it fits well with a review characterized by broad and explorative scopes [55,56].

This study was performed by adopting the methodological framework developed by Arksey e O’Malley [57] and advanced by Levac et al. [58], which entails the following steps: (a) identifying the research question, (b) searching for relevant studies, (c) selecting studies, (d) charting the data, and (e) collecting, summarizing, and reporting the results.

Finally, to ensure a systematic and consistent scoping review, data were reported according to the Preferred Reporting Items for Systematic Reviews and Meta-Analysis–Scoping Review (PRISMA-ScR) [56,59] checklist.

### 2.1. Research Questions

In order to map the state of evidence concerning how MOOCs are used in physicians’ education and to determine the extent to which they can be adopted in this field, this scoping review was conducted through a critical examination of topics that were not fully addressed formerly. Specifically, we aimed to answer the following research questions:What are the factors that shape the effectiveness of MOOCs aimed at physicians?Is MOOC a suitable teaching method for physicians’ training?

These questions were identified by using the PICO (population, intervention, comparison, outcome) framework [56]. More specifically, the population was defined as learners in the physician population and in all countries to ensure a better understanding of the state of evidence internationally. The intervention instrument evaluated was MOOC in medical education (including basic and advanced continuing professional development and university-level training). The comparison used in this review was the evaluation of the MOOC educational model in achieving learning outcomes compared to other educational methods, such as face-to-face learning. The outcome was the analysis of MOOC impact according to the four levels of training evaluation indicated by Kirkpatrick’s model [60]. As such, the latter was considered in terms of participants’ reactions, enhanced knowledge, changes in clinical practice, and further impact indicators considered by developers.

### 2.2. Search Strategy

Relevant articles were identified through three electronic databases: Scopus, PubMed, and PsycInfo. In order to determine the most suitable search strategy, three specialists in the fields of teaching, healthcare, and health psychology were consulted. Their suggestions were useful in identifying both the key search terms and databases most likely to produce the desired results. In addition to such a contribution, a preliminary analysis of the literature about MOOCs in healthcare contexts was performed with the aim of determining an adequate search strategy. Specifically, in line with previous studies on HCWs training [4], an algorithm of keywords was used. The latter was based on the following terms: “MOOC”, “Physicians”, “Doctors”, and “Massive Open Online Courses”; and Boolean operators: “AND”, “OR”, and “NOT”. The search timeline included articles published between 2016 and 2021. An example of a search string is as follows: Title-Abs-Key (MOOCs OR MOOC AND Doctors OR Doctor) AND (LIMIT-TO (Doctype, “ar”)) AND (LIMIT-TO (Pubyear, 2021) OR LIMIT-TO (Pubyear, 2020) OR LIMIT-TO (Pubyear, 2019) OR LIMIT-TO (Pubyear, 2018) OR LIMIT-TO (Pubyear, 2017) OR LIMIT-TO (Pubyear, 2016)) AND (LIMIT-TO (Language, “English”)).

### 2.3. Inclusion and Exclusion Criteria

The articles were selected on the basis of the following inclusion criteria: (1) qualitative and quantitative empirical studies regarding MOOCs for physicians, (2) written in English, (3) with full-text available online, (4) with clearly defined and explicit methods, and (5) published in peer-reviewed journals in the last five years (from 2016 to 2021).

Editorials, reviews, and dissertations were excluded, as well as studies summarizing the implementation of MOOCs, not including physicians, and published in languages other than English. No geographical restriction was used.

### 2.4. Charting the Studies

The search identified 112 studies for possible inclusion in the review. After 78 duplicates were removed, the titles and abstracts of 34 articles were screened for relevance independently by two reviewers, experts in training courses for HCWs and in work and organizational psychology. A total of 10 studies were excluded because they did not meet the inclusion criteria. An evaluation based on the full texts of the remaining 24 studies was carried out, leading to the exclusion of further 7 records. Hence, the final number of articles included in the scoping review was 17. The latters were subject to content analysis to identify potential categories based on the most prevalent themes and constructs across them. The reviewers discussed the differences or uncertainties regarding the inclusion of the studies during the content analysis phase. Figure 1 includes the flowchart of the literature search and screening process adopted in this review. The selected articles were charted into a table with the following information: authors; year of publication; country or countries; purpose; design; platform adopted; course duration; topic; health profession target; didactical, pedagogical, or research methods; evaluation system; and main findings. A brief description of the 17 articles selected is provided in Table 1.

## 3. Results

A total of 112 studies emerged, then 34 were screened and assessed for eligibility, leading to the exclusion of 17 papers. This resulted in a final corpus of 17 articles (see Figure 1 for the flowchart) published in a five-year time span from 2016 to 2021 (Table 1). More specifically, one article was published in 2016, three studies in 2017, five in 2018, 1 in 2019, two in 2020, and five in 2021. Moreover, the content analysis led us to include the articles in the following four main thematic areas: (1) pedagogical approaches, (2) MOOC structure-related variables, (3) participant-related variables, and (4) MOOCs vs. traditional courses.

### 3.1. Area 1: Pedagogical Approaches

Studies in this area [61,63,64] highlighted that the pedagogic approach takes on a major role in MOOCs aimed at physicians as well as other HCWs.

More specifically, Floss et al. [63] claimed the assumption that transformative learning should inspire MOOC activities. Thereby, the authors developed their course with the aim of integrating emotional, creative, and affective dimensions [79], following Mezirow’s description of the learning process as characterized by a sequence of emotions, critical self-reflection, and exploration of new roles able to result in knowledge and skills acquisition [80].

Magaña-Valladares et al. [64] suggested the effectiveness of the constructivist education model in the MOOC environment. Accordingly, the practice is regarded as crucial for significant learning achievements. This translated into a training program oriented toward the active involvement of participants who contributed to constructing their learning through various activities. According to the authors, the adoption of this educational model is fundamental to developing theoretical and practical competencies, since individuals’ experiences are reconsidered based on the new information obtained through the activities carried out in a specific context. In addition to the constructivism theory, Magaña-Valladares and colleagues [64] supported the pivotal role of the gamification approach in the MOOC environment. Gamification was considered a very effective method in engaging professionals—including physicians—with community responsibilities since such an educational approach represents a way to increase the attractiveness of learning content among students and foster their motivation to participate in the proposed activities. As a result, the authors developed a MOOC for HCWs characterized by a reward system. Specifically, the participants were rewarded for each completed activity, and their rankings were published online. This led to a competition between participants who could view their ranking and those of others.

The study by Blakemore et al. [61] documented the usefulness of Laurillard’s conversational framework [62] in the design of MOOCs. One of the main arguments of this approach is that the learning process should take the form of an interactive dialogue between teacher and learner through four phases that correspond to different learning experiences and media technologies: discursive, interactive, adaptive, and reflective [62]. Consequently, no one medium should be considered “perfect” to support the different aspects of learning. On the contrary, the adoption of various media is advisable to elicit them. Following this reasoning, in the MOOC developed by Blakemore and colleagues [61], the conversational framework concepts took the form of a series of digital health literacy activities (e.g., training on how to find health information online; critical evaluation of a range of online health-related resources; writing a brief; reviewing colleagues’ work).

### 3.2. Area 2: MOOC Structure-Related Variables

In this area, the works focused on variables related to MOOCs structures and development factors able to play a role in their effectiveness [65,66,67,68,69]: the platform adopted, testing MOOC outcomes with a pilot study, the reputation of MOOC developers, multimodal approaches, content, and certification.

The first key dimension is represented by the specific platform adopted to deliver MOOC content. Specifically, Jacquet et al. [65] pointed out the need to choose a platform carefully based on its accessibility and flexibility, mainly concerning the management of updates. As a result, these scholars used the edX platform, as it is characterized by easiness in updating MOOC content and structures, in addition to providing MOOCs both on computers and on smartphones through its mobile app.

Before making a MOOC available to the targeted population, a pilot version seems to represent a good strategy to collect valuable information on the potential of the course in achieving learning goals. More specifically, in this phase emerged the need to focus on participants’ experience with the structure and content of each module [66,68], such as the ease of use of the MOOC platform and the clarity of assessment questions. Evidence gathered by the pilot versions of a MOOC can, therefore, shape the structure and content of its final version and, in turn, contribute to improving its effectiveness.

In defining the structure and content of MOOCs, the engagement of professionals is essential. They should be chosen depending on their extensive experience and reputation in the field of study delivered through specific MOOCs [65,66,67,68]. In particular, their role emerges as fundamental in establishing the organization of MOOC modules and their timelines according to participants’ learning times [67].

Furthermore, in order to promote participants’ interest and engagement, the content of MOOCs was delivered through a multimodal approach by combining text, video, images, and audio files [65,66,67,69].

The above-mentioned aim was also achieved by using real examples and scenarios to illustrate common issues that participants might have encountered in their clinical practice [65,66]. Such a strategy represented a way to encourage the active participants of learners [66] as well as implementing a forum where they could discuss with peers and instructors [65,69]. A forum could also promote participants’ reflections on and sharing of what they have learned through the MOOC [69].

Finally, to obtain a reliable assessment of the effectiveness of their MOOCs, the authors mainly adopted quantitative questionnaires. These assessed participants’ knowledge [65,68,69], their understanding of the language adopted by MOOCs, and their degree of satisfaction [68], as well as their beliefs about courses’ effectiveness in providing career benefits, growing professional networks, and delivering useful content [66] that are usable in their clinical practice [69]. According to these assessments, a certification was offered if participants reached a certain percentage of module completion and knowledge, ascertained by online tests [69]. In this respect, Von Screeb et al. [68] decided to provide the mentioned certification at no cost since this might increase participants’ motivation and reduce dropouts.

### 3.3. Area 3: Participant-Related Variables

A total of six studies [70,71,72,73,74,75] were characterized by a focus on participant-related factors affecting MOOC outcomes: English proficiency, specialist knowledge, digital skills, previous experiences with MOOCs, lack of time, and motivation

In this respect, participants with low proficiency in English tend to face difficulties in attending a MOOC [70,71,74]. Such a barrier can translate into lower levels of participation and performance as well as higher dropout rates [71]. As argued by Medina-Presentado et al. [74], MOOCs developers must make specific choices to avoid such a factor creating an obstacle capable of undermining courses’ effectiveness in achieving learning goals. The authors built a MOOC on resistance to antibacterial drugs aimed at physicians and other healthcare professionals from Latin American countries. In order to reach as many participants as possible, the course was translated into the two main languages spoken across Latin America: Spanish and Portuguese. Translating MOOC content in the local language emerged as particularly helpful in those conditions where quick and effective physician training is needed (e.g., the COVID-19 emergency), as in the MOOC proposed by Findyatrini et al. [71]. The latter was characterized by literature reviews and instructions provided in the physicians’ native language to overcome the barrier constituted by participants’ low proficiency in English.

Henningsohn et al. [72] demonstrated that having specialist knowledge about MOOC content affects the achievement of learning objectives. The authors implemented the world’s first urology MOOC and evaluated the learning process according to two groups: specialist vs. non-specialist. Interestingly, no significant differences were reported, both in terms of completion rates and learners’ performances. Nevertheless, the non-specialist group required more time to complete the MOOC than the specialist group.

In addition to specialistic knowledge, a lack of digital skills and inadequate previous experiences with this teaching method [71,73] are variables capable of playing a role in MOOC effectiveness. A possible solution to this issue can be found in providing support to address the lack of technical skills and familiarity with MOOCs [71]. In this way, individuals are more likely to participate in MOOCs and express positive attitudes toward learning, especially when this support is comprehensive, convenient, and provided promptly.

Finally, the studies identified lack of time and low personal motivation as further participant-related variables to take into account in order to improve MOOCs efficacy [73,75]. In particular, physicians who could not dedicate as much time to attending the courses and responding adequately to the relative demands were revealed to be more likely to quit MOOCs. Likewise, participants with low motivation to enroll in MOOCs reported higher dropout rates [73]. These motivations mainly involve learning new things, specifically in HCWs’ clinical work. More specifically, doctors seem motivated by the possibility of engaging with university-level education and integrating their existing studies [75].

### 3.4. Area 4: MOOCs vs. Traditional Courses

Studies included in this area [76,77,78] compared the MOOC teaching model and the traditional educational ones.

The study by Brewer et al. [76] compared the learning outcomes of three modalities in teaching clinical examination: MOOC, face-to-face education, and a textbook chapter. The latter contained both written and pictorial descriptions of a standardized approach to the examination of shoulders, while the MOOC consisted of a video showing a physician teaching the same comprehensive examination described within the textbook; in the face-to-face teaching session, a physician taught a shoulder examination following the method adopted both in the textbook chapter and the custom video. Finally, the pre-clinical medical students who participated in the research underwent a standardized assessment of their competencies, reporting group-based differences. Specifically, participants in the face-to-face group showed higher scores than those in the MOOC group and the textbook group, leading to the conclusion that the face-to-face teaching mode represented the most effective method for teaching clinical examinations of the shoulder.

These findings might be explained by taking into account certain disadvantages of the MOOC model compared to traditional teaching that emerged from a series of qualitative evaluations on this teaching mode conducted by Pérez-Moreno et al. [77]. One of the mentioned dimensions was identified in the more restricted theoretical content that tends to characterize MOOCs, in contrast to the more extensive range of content available with traditional education methods. Moreover, the latters tends to be less plagued by the technical problems typical of online platforms such as those used by MOOCs.

The results mentioned above partially contradict those documented by Wang et al. [78]. These authors evaluated the effectiveness of two teaching methods on trainee physicians’ learning performances. Some of them enrolled in a traditional course, while others attended a hybrid course based on the MOOC model and adapted according to the flipped classroom approach. The latter entails that activities traditionally conducted in the classroom (e.g., content presentation) became home activities. In contrast, activities usually constituting homework become classroom activities (e.g., problem-solving and discussion) [81]. Participants in Wang and colleagues’ [78] study were placed in two groups: (1) traditional teaching and (2) MOOC plus flipped classroom. The study findings revealed that the physicians’ exam scores in the experimental group were significantly higher than those in the control group. In addition, most trainees reported being satisfied with the hybrid teaching model (MOOC plus flipped classroom), mainly due to the innovative thinking it promoted.

## 4. Discussion

The need to improve health organizations is widely acknowledged [1,43], as is the importance of well-trained healthcare workers to achieve this objective [31,82,83]. As argued above, the best strategy is represented by continuing education, which can improve HCWs’ knowledge, skills, resilience, performance, and, in turn, quality of care [2,84]. In order to combine the advantages of continuing education with the need to provide flexible, accessible, economically sustainable, and effective training, researchers and organizations have looked to MOOCs [85]. Regardless of this teaching model, a large amount of research [19] suggested that its effectiveness is based on a design built according to the job characteristics and personal needs of learners. Despite this evidence, there is a dearth of studies focusing on factors underlying MOOCs effectiveness among physicians. Consequently, the current study aimed to carry out a review of MOOC studies targeting physicians to extend the findings of the literature on this topic. Additionally, the review aimed to evaluate the potential of these courses to represent an adequate teaching model for such a population, as well as the factors able to explain their learning-related outcomes.

The analysis identified 17 studies that met the predefined inclusion criteria. The paper provided an overview of these studies, considering the context, the participants, the methodology adopted, and the main results. In addition, four areas emerged from the content analysis of the studies examined: pedagogical approaches, MOOC structure-related variables, participant-related variables, and MOOCs vs. traditional courses.

Concerning the review findings, the first evidence consists of the crucial role played by defining which pedagogical approach to adopt in building a MOOC. More specifically, it was suggested that MOOCs for physicians, designed with an adequate pedagogical foundation, could improve learning outcomes, in line with further studies [5]. Among the mentioned pedagogical theories, transformative learning, constructivism, gamification, and the conversational framework came to light. This result demonstrates two different characteristics of MOOCs aimed at physicians. On the one hand, it refers to the great flexibility of this teaching mode, as it can be based on different educational approaches depending on the specific content and goals of the courses [86]. On the other hand, only a few studies reported that a psychological approach was adopted, suggesting a potential weakness of these online courses consistent with the low amount of evidence and agreement in defining the characteristics of the MOOC model, as well as its pedagogical foundations in other populations [5]. However, despite the differences between the pedagogical theories reported, it is possible to identify a recurrent emphasis on the role of practice in gaining new knowledge and skills. The literature highlights that promoting learning-by-doing education is crucial in learning in various educational contexts [87], including MOOCs. In fact, higher information retention rates and better learning outcomes are achieved when learners apply their theoretical knowledge to practical endeavors [88,89].

The relatively small number of studies reporting their pedagogical foundation is partially counterbalanced by several characteristics that emerged in the analyzed MOOCs, which can be easily traced back to the main assumptions of the aforementioned pedagogical approaches. This reasoning is supported, for example, by the choice made by several authors [65,67,68,71,74,75,78] to use a range of teaching techniques (e.g., text, video audio, or case studies) in order to consider various participants’ learning modes, as well the different dimensions of learning. Such a strategy follows the focus on experience in the learning process, as highlighted by the above-mentioned pedagogical dimensions [90]. In particular, the use of interactive case scenarios, which physicians may experience in their clinical practice, was proven to foster their active participation and, in turn, their engagement and learning outcomes [65,66]. In this way, it might be possible to avoid the passiveness of participants, which is identified as one of the main risk factors for MOOCs dropout [50].

Following the same line of reasoning, additional MOOC features were implemented to motivate and enhance participants’ learning: online quizzes, peer assessments, and discussion forums. As reported by studies included in this review [65,69], discussions through forums or social networks (e.g., Facebook) provide interactive spaces where physicians can share their reflections and issues related to courses with peers and instructors, in addition to enhancing their professional networks. This thesis is supported by a large amount of research demonstrating that learners who participate in forums have higher MOOC pass rates [50] than those who are inactive in online discussions. Moreover, as reported in Jacquet and colleagues’ [65] study, including a moderator in these platforms is crucial to support their usefulness in the learning process, as is consistent with the literature [91] highlighting how e-moderation in the MOOC environment allows for the establishment of a setting with individualized support for students, which is able to promote interaction and collaboration and, consequently, the construction of knowledge.

A further pivotal dimension of MOOCs that emerged in the examined studies refers to the content taught. In particular, it underlined the need to select content based on its usefulness and appropriateness to the physicians’ clinical practice [66,69]. The literature has shown that perceived usefulness positively impacts learners’ attitudes and satisfaction with training courses, as well as motivation; this, in turn, might improve learning outcomes [92]. Several studies [93] explain this link by adopting the theory of planned behavior [94]. More specifically, researchers demonstrated that the perceived usefulness and ease of use of a MOOC are related to students’ attitudes toward the course and their intention to attend it. Therefore, it is necessary to pay attention to specific needs and learning styles that can influence MOOCs effectiveness among physicians. In this regard, Pickering and Swinnerton’s [75] findings highlighted the need to develop content to satisfy physicians’ demand to be engaged with university-level education and to integrate their previous studies, with the aim of preventing dropouts [73].

Furthermore, a lack of time and low personal motivation are factors that may negatively affect MOOCs effectiveness [73,75]. Physicians who could not dedicate as much time to attending the courses and could not respond adequately to the relative demands were more likely to quit MOOCs. Likewise, participants with low motivation to enroll in MOOCs showed higher dropout rates [73]. This means that identifying a flexible teaching structure compatible with physicians’ time [67] is fundamental in the MOOC environment. This is related to the limited time resources of the physicians’ job, since it is often constituted by full-time employment [95]. Additionally, to increase courses’ accessibility and their outcomes, action needs to be taken on barriers related to lack of knowledge among learners. In particular, inadequate English language proficiency, low knowledge about MOOC content and structure, and a lack of digital skills [70,71,72,74], together with low motivation [75], were identified as risk factors for poor performance, low levels of physician engagement, and high dropout rates. Consequently, it is essential to personalize course structure, content, and activities according to its target [20,75,96]. This entails collecting information about the target population (e.g., participants’ skills, knowledge already acquired on the MOOC content and that obtained by attending a MOOC, their socio-demographic information, job role, and clinical specialization) during the pilot phase [66] or the implementation phase [67,68,69]. This information is detected before the first module of a MOOC or once it is completed and contributes to defining an appropriate assessment tool that includes both questions related to the MOOC learning aims (e.g., improved knowledge or intention to change their clinical practice according to MOOC content) [68] and to the experience of learners with the specific course (e.g., course satisfaction, content clarity) [67]. The most common methodology assessment used in the examined studies is the quantitative one [65], presumably adopted due to the large number of MOOC participants. Such a methodology translates into the use of automated quizzes, functional both for instructors (to track trainees’ progress and MOOC success) and for learners (to receive feedback on their responses) [19].

It must be noted that implementing an appropriate assessment tool and the further characteristics mentioned above requires a suitable platform to host MOOCs [65]. The latter should be selected on the basis of the compatibility of its features (e.g., update management, flexibility) with the course aims, structure, and content. Moreover, this requires the engagement of a team of professionals recruited according to their extensive experience and reputation in the field of study of the specific MOOC [65,66,67,68]. In support of this, several studies recognized the reputation of MOOCs developers as a strong predictor for selecting a certain MOOC [97] since it is considered a manifestation of high-quality content [98].

Finally, in comparing MOOCs and traditional courses, previous studies reported no significant differences [24] or better outcomes in favor of the MOOC model [23]. Surprisingly, the studies included in the current review seem to partially challenge these findings. More precisely, Brewer et al. [76] documented better learning outcomes in the experimental group with traditional face-to-face training than in the MOOC group. However, in contrast to this result, Wang et al. [78] demonstrated the potential of the MOOC model to lead to higher learning performance and satisfaction levels than traditional education when it is combined with the flipped classroom approach. Taken together, these findings suggest investigating further factors able to shape the learning outcomes of MOOCs. In line with Alario-Hoyos et al. [99], it is plausible to suppose that an explanation for the findings of Brewer et al. [76] might be found in the characteristics adopted to build their MOOC. It was developed through a mere replication of a textbook diagnostic chapter with no care for the distinctive aspects of MOOC “language”, such as its multimodal or interactive dimensions, as previously mentioned. For example, no interaction between physicians and teachers was possible in the MOOC group designed by the authors.

## 5. Conclusions

The rise of the MOOC model in the context of teaching and learning has impacted the healthcare field due to its potential to offer a flexible, asynchronous, timely, and cost-effective mode of training [42], which is recognized as pivotal in improving the efficacy and performance of healthcare organizations and, in turn, the quality of care [2].

Regarding the best strategies for developing MOOCs for physicians’ training, the papers included in the review reported the need to adopt an approach based on experiential learning and, in turn, to define the design and structure of MOOCs. As a result, implementing interactive activities is crucial to improve students’ satisfaction, engagement, and acquisition of skills and knowledge, as well as the adoption of MOOC content in their clinical practice.

It is noteworthy that the aforementioned dimensions regarding the pedagogical foundation and design of MOOCs must be read in conjunction with participant-related variables (i.e., medical knowledge, digital skills, previous experiences with MOOCs, lack of time, and motivation) that are able to shape the impact of the former and improve their effectiveness.

Finally, the MOOC model emerged as a functional training method to ensure valuable training to physicians, even compared to traditional education, on the condition that developers do not adopt passive instructions, which might jeopardize their potential in the learning process.

We are aware that this review has some limitations that should be considered. First, this study used only a few databases to search for empirical MOOC studies; hence, there might be some studies that were not included in these directories. Likewise, their exclusion could have been determined by the inclusion criteria used (e.g., English language). Moreover, although a systematic approach was used, some information bias could have affected both the studies examined and the findings. An additional limitation regards information about MOOC characteristics that were not always available since various research designs were adopted by the studies included in the review. Additionally, undefined concepts (e.g., “learning outcomes”) often emerged in the studies, suggesting the need to improve the operationalization of the concepts adopted in data reported within this field of research, as documented by Raffaghelli et al. [100]. A further weakness consists in the prevalence of the cross-sectional studies and self-reported measures that, with the differences in participants’ characteristics, limit the generalizability of our findings.

Even though this explorative review may not represent a complete picture of the MOOC phenomenon, it merits attention because it sheds light on the variables related to MOOCs effectiveness for the physician population. Moreover, the present study extended the findings of previous research analyses by including more recently published MOOC studies to detect innovative teaching strategies related to the MOOC educational context, which is strongly affected by the development of new technologies.

To summarize, the findings of this review support the thesis that the MOOC model can be conceived as a viable resource for training physicians due to its flexibility in adopting a range of pedagogical frameworks and innovative teaching and learning strategies. Moreover, it can be argued that various and connected dimensions contribute to MOOCs effectiveness. Specifically, building a MOOC for physicians entails considering a range of variables related to learners’ characteristics, course content, and the course structure, which are a direct manifestation of the pedagogical approach adopted. In this regard, this paper offers valuable support to educational and healthcare organizations aiming to gain appropriate knowledge of the target physician population to foster engagement and, in turn, achieve better performance and improve the quality of care.

Finally, since this research field only partially explored MOOCs aimed at physicians, future studies should focus on the role of further strategies that are useful in enhancing physicians’ engagement, such as the role of more extensive adoption of new social network platforms or virtual reality to pursue a twofold goal: to promote participants’ engagement and to minimize the dropout phenomenon.

## Figures and Tables

**Figure 1 ijerph-19-14247-f001:**
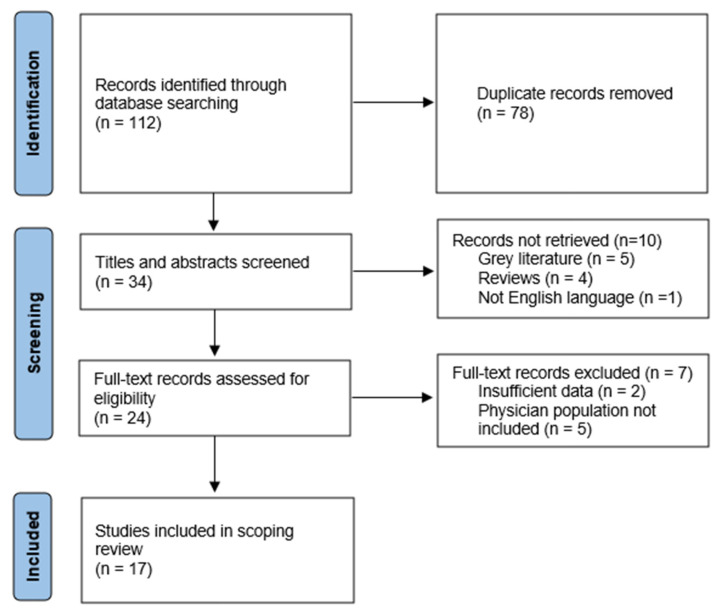
Flowchart of the literature search strategy and review process [59] (adapted from PRISMA 2020).

**Table 1 ijerph-19-14247-t001:** Summary of the articles included in the scoping review.

Authors (Year) Country Dev.	Study Aims; Study Design	Platform; Course Duration	Topic; Health Profession(s)	Methods	Assessment Methods	Results
Area 1: Pedagogical approaches						
Blakemore et al., 2020 [61] UK	To assess the effectiveness of a series of educational interventions in improving the ability of learners to evaluate online health care information; Iterative mixed methods study	FutureLearn Six weeks	Cancer genomics; Various	Content (videos, articles, podcasts, discussion forums, Twitter chats, small group projects, written assignments, peer reviews) inspired by Laurillard’s conversational framework [62]	Evaluating the number of references included for each final course assignment; peer review evaluation of learners’ assignments	Statistically significant increase in the number of references cited in the final written assignments
Floss et al., 2021 [63] Brazil	To evaluate the efficacy of a MOOC on the climate crisis; Descriptive study adopting a mixed methods approach	TelessaúdeRS-UFRGS 80 h	Planetary health; Professionals working in primary health care	Content (articles, short videos, podcasts) inspired by transformative learning	Assessment through multiple choice questionnaires before and at the end of the course; assessment after the completion of each module. Qualitative analysis through forum participation; submitted action plans; videos talking about learners’ experience	High completion rates; most participants reported being very satisfied with the learning experience
Magaña-Valladares et al., 2018 [64] Mexico	To document the key success factors in improving HCWs’ competencies through implementing a MOOC; Pre-post	Moodle 40 h	Breast cancer; Primary HCWs (physicians, nurses, health promoters, and medical students)	Content (videos and interactive exercises adapted to the specific learners’ group) inspired by constructivism and gamification theories	Pre- and post-tests adopted for learning assessment; evaluation at the end of each module	Graduation rates higher than those previously reported in the literature
Area 2: MOOC structure-related variables						
Jaquet et al., 2018 [65] USA	To investigate the effectiveness of a MOOC in training learners to engage in safe and ethical global health experiences Pre-post	edX; 3 weeks	Safety and ethical aspects in global health experiences; Medical students	Texts; interactive case studies; documentary-style video narratives; photographic images; practice questions; reflective exercises; discussion forum	Pre- and post-tests, consisting of multiple-choices questionnaires, adopted for learning assessment; evaluation at the end of each module	Mean post-test scores significantly improved in a subsample
Lunde et al., 2018 [66] NR	To investigate the efficacy of a pilot MOOC in introducing and promoting clinician skill development among healthcare workers; Case study	NR NR	Health assessment and clinical in primary care; Physicians; nurse practitioners, nurses; nurses aid; students in healthcare education (medical students; master and bachelor students in nursing)	Videos; audio files; texts; realistic examples and scenarios	Evaluation of satisfaction with each module	The modules were perceived as useful and relevant
Nampoothiri et al., 2021 [67] UK	To describe how the use of a MOOC provided an opportunity to gather diverse narratives on AMS from around the world Explorative study	NR Three weeks	Antimicrobial stewardship Doctors; pharmacists; nurses; students (mainly medical, nursing and pharmacy); researchers	Video case presentations	Knowledge tests for each module	NR
Von Schreeb et al., 2020 [68] Various	To evaluate the knowledge improvement from a MOOC, the influence of language, course satisfaction, and the subsequent effect on the intention to change antibiotic prescribing behaviors; Pre-post	Coursera Four weeks	Antimicrobial Stewardship Physicians	Videos with experts in the field	Four surveys: two knowledge assessments (using a multiple choices questionnaire), one satisfaction survey, and a CME test	Significant improvement in knowledge, mainly among participants with a higher English level; physicians reported high or very high intentions to change their clinical practice
Wewer Albrechtsen et al., 2017 [69] Denmark	To evaluate whether participation in a MOOC in the prevention and treatment of diabetes and obesity had any impact on the knowledge, skills, and career of healthcare professionals by comparing outcomes of participants from developing countries and those from developed countries; Explorative	Coursera Six weeks	Diabetes; Medical doctors; nurses; midwives, nutritionists; technicians working in healthcare	Reading materials; video lectures; discussion forum with peers and the instructor	Multiple choice questionnaire; survey with nine closed-ended questions	Improvement in subjective knowledge (objective knowledge assessed but not reported) mainly by HCWs from developing countries compared to those from developed regions
Area 3: Participant-related variables						
Charani et al., 2019 [70] NR	To describe the findings of a global survey of HCWs on the implementation of antibiotic stewardship programs and report the motivations for the prescribing of antibiotics; Online survey	FutureLearn; Five weeks	Antibiotic stewardship; Doctors; nurses; pharmacists	Learning materials; discussion forum	Voluntary self-assessment through a questionnaire including questions about demographic data, stewardship activities, motivations for antibiotic prescribing	Factors influencing antibiotic prescribing were consistent across all healthcare professions.
Findyartini et al., 2021 [71] Indonesia	To analyze and evaluate participants’ satisfaction and increase in knowledge after completing a MOOC; Pre-post	Moodle; 42 h	COVID-19 management; Newly graduated medical doctors	Video; text/embedded guidelines/slides; podcast; references	Qualitative assessment performed by adopting the Quality Reference Framework checklist; quantitative assessments carried out through pre- and post-tests about knowledge gained; evaluation of the participants’ satisfaction assessed both quantitatively and qualitatively through seven open-ended questions.	Most participants thought that the platform was easy to navigate, the design was interesting, and that the content was aligned with their needs; pre- and post-test scores increased significantly
Henningsohn et al., 2017 [72] Sweden	To report the design and outcomes of a urology MOOC and outline the differences between specialist and non-specialist participants; Case study	NR; NR	Urological diseases; Various	Video lectures; FAQ; branched virtual patients with continuous feedback; glossary; three-dimensional models of the genitourinary tract	Case-based multiple-choice questions after each video; qualitative assessment to evaluate the course	Most participants reported having achieved the learning outcomes for the course to a large or very large extent; the non-specialist group spent more time on the learning material than the specialistic one
Hoedebecke et al., 2018 [73] USA	To describe the experience of young family doctors attending a MOOC; Explorative	EdX; Facebook; WhatsApp; Google Documents; NR	Research in Primary Healthcare; Family doctors; resident doctors	Didactic lecture videos from primary care research experts; peer-to-peer interaction; mentoring; a mental map created by one participant to summarize the learned content once each module was completed; weekly reminders; updates through social media	Each module concludes with a quiz to evaluate knowledge learning and retention	A high completion rate was indicated; further information assessed was not reported
Medina-Presentado et al., 2017 [74] NR	To describe the design, implementation, and results of continuing interprofessional education program on hospital-acquired infections and antimicrobial resistance in Latin America; Pre-post	EviMed; Eight weeks	Hospital-acquired infections and antimicrobial resistance; Physicians; microbiologists; infection control professionals	Individual clinical cases; reading materials; discussion forum; videos and voice-over-presentations; patient aids; electronic rounds (e-rounds) on clinical cases; clinical simulations	Pre- and post-tests about participation, satisfaction, and knowledge gained	Significant increase in knowledge between before and after the course, mainly among participants who took both tests; most respondents who completed the satisfaction survey defined the course as “very good.”
Pickering and Swinnerton, 2017 [75] UK	To evaluate the use of an anatomy MOOC as part of a blended learning medical anatomy curriculum and provide valuable information about the demographic profile, patterns of engagement, and self-perceived benefits linked to the course; Explorative	FutureLearn; Three weeks	Clinical anatomy of the abdomen; Doctors; nurses; allied health professionals	Video lectures; discussion forum	Quantitative and qualitative assessment through automated self-assessment 21 questions survey about demographic data, motivation, engagement, and self-perceived benefits	Most participants followed the suggested course pathway and considered the MOOC a useful tool compared to other resources available; differences in motivation for enrolling in the MOOC were reported
Area 4: MOOCs vs. traditional courses						
Brewer et al., 2021 [76] UK	To determine whether three teaching modalities were of equal efficacy in medical students’ training; Prospective cohort study	SproutVideo; 21 days	Shoulder examination; Pre-clinical medical students	Three groups: textbook study, face-to-face seminar, and video tutorial	Assessment of knowledge before and after the course; learning style questionnaire	Higher scores in the face-to-face group followed by video tutorial group and textbook; no influence of learning style on scores
Pérez-Moreno et al., 2018 [77] Spain	To evaluate the impact of a MOOC on the appropriate use of antimicrobial agents and determine specific study areas with better learning outcomes; to identify course weaknesses; Pre-post	Iavante; Four months	Use of antimicrobials in major syndromes of serious infectious diseases; Physicians; pharmacists	NR	Voluntary self-assessment with 30 questions using a quantitative approach; an open response section to evaluate participants’ satisfaction	A significant increase in self-assessment for all the questions; differences between the MOOC and a traditional course were detected
Wang et al., 2021 [78] China	To evaluate the feasibility, acceptability, and effectiveness of MOOCs in combination with flipped classroom teaching in the standard training of resident physicians; Prospective cohort study	NR; NR	Rheumatoid arthritis; Resident physicians	For the experimental group only: short videos watched before class and intra-group discussions; clinical cases	A quiz about learning content; an evaluation survey about the teaching method	Significant differences in the exam scores in favor of the experimental group

Note: Country dev. = country where the course had been developed; HCWs = Healthcare workers; NR = non reported; h = hours; AMS = antimicrobial resistance; CME = continuing medical education.

## Data Availability

Not applicable.

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
