# Peer review of "Learning Design Strategies in MOOCs for Physicians’ Training: A Scoping Review"

_ijerph, 2022, doi:10.3390/ijerph192114247_

Round 1
Reviewer 1 Report
First of all, I would like to thank the authors for this interesting work. Many of the strengths of this research, including the scientific design according to the PRISMA approach and using Inclusion and exclusion criteria. However, there are several essential notes that could be consider for the manuscript to be publishable.
Firstly: the title should be shorter, and I suggest Learning Design Strategies in MOOCs for Physicians’ Training: A Scoping Review. Moreover, in the abstract the authors should Define Acronym before using it e.g. In recent years, the increasing implementation of massive open online courses (MOOCs) has emerged.
Secondly: The authors' use of (6) references to support one sentence, “Literature has demonstrated that the adoption of human resource development in terventions can be helpful in increasing healthcare workers’ (HCWs) knowledge, skills, resilience, and, in turn, quality of care [2–7]” which I think is exaggerated, and only one reference is sufficient, and the others should exclude. The same for this sentence “These findings are aligned with the Human capital theory [8–10]”
Thirdly: The introduction is too long. It should be reorganized as suggested into two parts. The first is an introduction to the motivation for conducting the research, its objectives, and the need for a procedure to support physicians’ training, and the second part is for related studies.
Fourth: In the conclusion the authors used phrases directly from the manuscript with its references. This is not acceptable in the formulation of the conclusion. The conclusion of a research paper is where you wrap up your ideas and remind the reader why the work presented in the paper is relevant. In the conclusion section, I suggest that repeat the research goal, method, and the major findings of your study. then the authors should move from specific findings or arguments back to a more general depiction of how the manuscript contributes to the readers’ understanding of a certain concept or helps solve a practical problem or fills an important gap in the literature. The content of the conclusion section therefore depends on the type of research you are doing and what type of paper you are writing e.g., a scoping review. The conclusion also is where the authors briefly summarize it and place it within a larger context. It could be called the “take-home message” of the entire paper.
Fifth: Some important references should be used especially as a motivation in this research and I strongly recommended them:
Moore, R. L., & Blackmon, S. J. (2022). From the Learner's perspective: A systematic review of MOOC learner experiences (2008–2021). Computers & Education, 104596.
Yousef, A. M. F., & Sumner, T. (2021). Reflections on the last decade of MOOC research. Computer Applications in Engineering Education, 29(4), 648-665.
Liu, C., Zou, D., Chen, X., Xie, H., & Chan, W. H. (2021). A bibliometric review on latent topics and trends of the empirical MOOC literature (2008–2019). Asia Pacific Education Review, 22(3), 515-534.
Finally, I think that there is no need for this sentence “Regarding the teaching methods to implement such a strategy, it must be noted that 47 healthcare has been marked by the rise in technology-enhanced teaching [19,20] fostered 48 by steadily growing internet access and the wide adoption of web-based platforms in in-49 dividual lives [21,22].” As it includes 3 references to the same authors, and thus the number of references for self-citation is (4), which is a large number. This should be reduced to a maximum number of (2) self-citation.
Author Response
Thank you for the feedback on our manuscript. The latter has been revised according to the suggestions. Below are the responses to the comments. We hope that we have answered satisfactorily.
Please see the attachment.

Reviewer 2 Report
Thank you for the opportunity to review this article. Scientific and technological innovation is in continuous development, so it is important to know the constraints and advantages of implementing all available resources. In this case, teaching is no exception. I consider the topic very current and relevant.
I remind you that all the suggestions provided in this review are suggestions for improvement and with the aim of promoting reflection around the analyzed object.
Title:
It reflects the content covered and the type of study carried out. It is properly worded.
Summary:
The authors emphasize the essential points to understand the study, namely the justification of the theme, objective of the study, type of study, as well as the main results
Introduction:
The main concepts under study are presented, as well as the relationship between them. The authors also substantiate the relevance of the present study, based on relevant studies on the subject. It would be important to mention whether the authors registered the protocol of this Scoping Review. How did the authors guarantee the non-existence of any existing review on this topic?
Methods:
The authors used the PICO strategy. Since this is a Scoping Review, wouldn't it be more appropriate to use the PCC strategy? Since they intend to assess effectiveness, wouldn't it be more appropriate to carry out a systematic literature review? Since these are quantifiable data, why did the authors not carry out a meta-analysis? Did the authors carry out any preliminary research to define the keywords? How did you define your search terms? The authors report that they did research on Scopus, PubMed and PsycInfo, however, they do not present the search strategy. Were the search terms suitable for each of the databases?
Since this is a Scoping Review, wouldn't it be appropriate to include literature reviews and dissertations? Why did you choose not to include these documents?
The data extraction table includes the aspects necessary to meet the objective of the study. Is the comparison between MOOCs versus traditional courses a topic/category? Authors should clarify this aspect throughout the document, as it is not easily understood by readers. I suggest that you present the themes/categories before the presentation of table 1.
Results:
The results are presented adequately for the readers to understand, and reflect the content of a Scoping Review, however, in the methods this option is not clear.
Discussion:
The authors compare their results with the most current and pertinent evidence available and recognize the main limitations of the present study, stressing the need to invest in primary studies on the subject.
Conclusion:
The conclusion is very extensive and the authors refer again to bibliographic references. This section should be succinct and reflect the objective-oriented results of the study.
The authors identify the main limitations of the study.
Bibliography:
Authors should limit bibliographic references to the necessary number, and should only include current sources.
Author Response
Thank you for the review of our paper, which has been improved based on the suggestions. We hope we have adequately answered the reviewer’s remarks.
Please see the attachment.

Round 2
Reviewer 1 Report
Many thanks to the authors for the new edits. Excellent manuscript, well worth publishing.
Reviewer 2 Report
ongratulations to the authors,
The manuscript is of publication quality. However, I suggest that in future submissions you record all the changes made, namely in the bibliographic references, since it facilitates the work of the reviewers.